# Targeted RNAseq Improves Clinical Diagnosis of Very Early-Onset Pediatric Immune Dysregulation

**DOI:** 10.3390/jpm12060919

**Published:** 2022-06-01

**Authors:** Kiera Berger, Dalia Arafat, Shanmuganathan Chandrakasan, Scott B. Snapper, Greg Gibson

**Affiliations:** 1School of Biological Sciences, Georgia Institute of Technology, Atlanta, GA 30332, USA; kberger9@gatech.edu (K.B.); dalia.arafat.gulick@emory.edu (D.A.); 2Division of Bone Marrow Transplant, Children’s Healthcare of Atlanta, Emory University School of Medicine, Atlanta, GA 30322, USA; shanmuganathan.chandrakasan@emory.edu; 3Division of Gastroenterology, Hepatology, and Nutrition, Boston Children’s Hospital, Boston, MA 02115, USA; scott.snapper@childrens.harvard.edu

**Keywords:** RNAseq, very early-onset inflammatory bowel disease, primary immune deficiencies, splicing, rare disease

## Abstract

Despite increased use of whole exome sequencing (WES) for the clinical analysis of rare disease, overall diagnostic yield for most disorders hovers around 30%. Previous studies of mRNA have succeeded in increasing diagnoses for clearly defined disorders of monogenic inheritance. We asked if targeted RNA sequencing could provide similar benefits for primary immunodeficiencies (PIDs) and very early-onset inflammatory bowel disease (VEOIBD), both of which are difficult to diagnose due to high heterogeneity and variable severity. We performed targeted RNA sequencing of a panel of 260 immune-related genes for a cohort of 13 patients (seven suspected PID cases and six VEOIBD) and analyzed variants, splicing, and exon usage. Exonic variants were identified in seven cases, some of which had been previously prioritized by exome sequencing. For four cases, allele specific expression or lack thereof provided additional insights into possible disease mechanisms. In addition, we identified five instances of aberrant splicing associated with four variants. Three of these variants had been previously classified as benign in ClinVar based on population frequency. Digenic or oligogenic inheritance is suggested for at least two patients. In addition to validating the use of targeted RNA sequencing, our results show that rare disease research will benefit from incorporating contributing genetic factors into the diagnostic approach.

## 1. Introduction

Under the general umbrella of personalized medicine, precision genomic medicine refers to investigations designed to diagnose the molecular cause of a clinical condition [1]. Whereas large biobank projects such as the TOPMed Precision Medicine Program [2] focus on gathering genomic and phenotypic data to elucidate patterns in populations that will allow researchers to develop risk predictions, clinicians must be able to design therapeutic interventions tailored to the individual’s genetics. As sequencing costs have come down, the focus has shifted from individual diagnostic odysseys involving trains of tests, to whole exome and genome sequencing [3]. Such studies of single individuals or families have typical diagnostic yields between 30% and 50% of patients, in many cases leading to more actionable information than that available from phenotypic considerations alone [3,4,5,6,7,8,9,10,11,12,13]. Precision medicine is also often lauded as the path towards targeted therapies for rare diseases that have previously not received much research funding or attention. Although the vast majority of success stories in precision medicine-based therapies are related to cancer [14,15], inherited immune disorder research is also beginning to yield a number of successful targeted therapies as well [16]. Here, we ask whether complementation of exome sequencing with targeted RNA sequencing can increase diagnostic yield in this context.

Primary immunodeficiencies (PIDs), also termed inborn errors of immunity (IEIs), encompass over 400 distinct disorders related to immune dysregulation [17,18,19,20]. This includes susceptibility to infection or malignancy, autoimmune and autoinflammatory disorders, and allergies. PIDs have historically been branded as monogenic disorders with traditional Mendelian inheritance [21], but as more PIDs have been identified and we have gained better understanding of immune function and clinical pathogenesis, it has been recognized that these disorders often display variable penetrance and severity [22]. The term complex immune dysregulation syndrome captures the idea that many cases of aberrant immune activity that share similar presentation nevertheless may have heterogeneous, and sometimes oligogenic, causes [23].

Adult-onset inflammatory bowel disease (IBD) is known to be a complex disease involving multiple genetic and environmental factors that leads to over-activation of the inflammatory response in the gastrointestinal tract [24,25]. Pediatric IBD makes up a quarter of all diagnosed IBD, and a subset of these cases occur in patients <6 years of age [26]. These are termed very early-onset IBD (VEOIBD) and are generally thought to have a simpler genetic basis [27,28,29]. VEOIBD has also been associated with PIDs, both through symptoms as well as gene involvement [23,28,30,31,32]. Previous exome studies of patients with VEOIBD have identified monogenic causes for a small percentage, but most cases remain without a genetic diagnosis. Monogenic cases of VEOIBD are more likely to have family history of IBD or immunodeficiencies and to be more severe and resistant to conventional treatment [27,28]. Research has also found patients with VEOIBD to have a higher rate of variants in genes associated with PIDs [31], suggesting that some cases for which a monogenic origin is not identified may have a multigenic etiology.

For both PIDs and VEOIBD, identifying the specific underlying genomic cause is important for treatment [21,23,30,33]. Gene-specific targeted therapies enable improved patient management and have been successfully used for several immune conditions [1,18,34]. This group of highly heterogenous disorders often exhibit cascading effects where multiple genes or pathways are pathogenically altered. In some cases, targeting an affected but not causal gene results in worse patient outcomes [23]. Since diagnostic yield from exome sequencing remains well under 50%, the accessibility of immune cells for genomic profiling of peripheral blood samples obtained by standard phlebotomy raises prospects of RNA-based analyses, specifically RNAseq, that might identify aberrant molecular events such as altered splicing or gene expression [5,35,36,37,38,39]. The relevant mutations might not be observable in exome sequences, or may be of uncertain significance.

Previous cohorts of WES for the diagnosis of PIDs and VEOIBD have prioritized genes known to be associated with immune disorders [21,27,31]. Recent studies have shown that periodic reanalysis of variants results in additional diagnoses [40,41,42,43,44]. This is lauded as a benefit of whole genome (WGS) and whole exome (WES) sequencing, allowing for genes newly associated with a disease to be reconsidered for patients. While most WES and WGS reanalysis reports an increase in diagnostic yield of around 12% with the majority coming from new gene–disease associations, numerous studies have found that for a given disease, patients harbor causative variants in genes that have already been identified for that disorder or group of disorders [45,46]. Novel disease gene discoveries are still happening at a high rate, but there is still a high burden of variants of uncertain significance (VUSs) and little incentive to systematically resolve them [47]. Low cost and high read depth motivate the development of targeted panels, which have been used at the DNA level in clinics for several decades now, but are just beginning to be considered for RNA analyses.

RNAseq complements WES and WGS by providing evidence of mRNA effects (or lack thereof) for specific variants as well as identifying alterations in splicing or other structural changes that DNA sequencing methods cannot see [37]. In large cohorts, transcript abundance can also be used to analyze downstream effects of some pathogenic variants and characterize pathways involved in disease [48,49]. Even where a likely pathogenic variant is identified by DNA analysis, RNAseq can provide supporting evidence that the transcript is affected, and may be used to establish that both alleles are affected in trans by different mutations [50,51]. Although peripheral blood is a mixture of dozens of cell types, so long as the defect is observed in one or more of the major leukocyte or monocyte populations, bulk RNAseq should be a useful source of clinically actionable information. In this study, we show that targeted RNAseq for a set of 13 PID and IBD patients resolves the likely source of immune dysfunction for at least eight patients, where previous exome analysis had only diagnosed three cases.

## 2. Materials and Methods

Targeted RNAseq was performed on samples from 13 patients known to have or suspected of having very early-onset inflammatory bowel disease (VEOIBD; 6 samples) or primary immune deficiencies (PID; 7 samples). Results of whole exome sequencing (WES), performed previously on all 13 samples, were withheld until initial analysis of RNAseq was completed to compare the success of RNAseq analysis without supplemental genome information.

### 2.1. Sequencing

This study used RNA (median RIN = 9.4, range = 6.9–9.8) from peripheral blood monocytes (PBMCs) as the sample type. PBMCs were extracted from whole blood using Stem cell Technologies Lymphoprep Density Gradient Medium. For our PID samples, RNA was extracted from PBMCs and sent to us directly from Emory University. For our VEOIBD samples, RNA was extracted from PBMCs using the NucleoSpin RNA XS kit (Macherey-Nagel, Düren, Germany). For targeted RNAseq strand-specific library preparation, RNA input per sample was 300 ng with target enrichment using a custom Agilent SureSelect Targeted Enrichment panel of 260 genes linked to VEOIBD or PID. All samples underwent Poly-A selection, followed by 150 bp paired-end sequencing on an Illumina NextSeq platform performed by Georgia Tech’s High Throughput DNA Sequencing Core at an average sequencing depth of 11.5 M read pairs. Further details of library preparation can be found in the SureSelect^XT^ RNA Target Enrichment System protocol (https://www.agilent.com/cs/library/usermanuals/Public/G9691-90000.pdf, Accessed on 26 July 2019).

### 2.2. Alignment and Pre-Processing

After QC of the raw sequencing files using FASTQC [52], reads were aligned to GRCh38 using GENCODE v29 [53] with STAR splice aware aligner version v2.6.1d [54]. In addition to default 2-pass mode parameters, all multimapping reads and splice junctions with <5 supporting reads were filtered out. Post-alignment QC was performed using the Quality of RNAseq Toolset (QoRTs) [55], primarily checking gene-body coverage plots for signs of 5′ or 3′ bias that may affect downstream analysis.

### 2.3. Variant Calling

Variant calls were made on aligned BAM files following GATK Best Practices for Variant Calling in RNAseq [56]. A single VCF file for all samples was created using GenotypeGVCFs and standard filters were applied. Variants were annotated using ANNOVAR [57] and TraP [58] and subsequently prioritized to identify high-quality (ALT read depth ≥10 unique non-duplicate reads) exonic variants and variants that may affect splicing. Notably, all three variants (two intronic, one exonic) found to affect splicing had TrAP scores well above the “probably damaging” threshold. The quality of all variants reported were manually confirmed using IGV viewer [59].

### 2.4. Allele Expression Imbalance

The presence of nonsense-mediated decay (nmd) acting on transcripts with nonsense variants was determined using a binomial probability test. To roughly account for possible mapping bias towards the reference allele, a null hypothesis of [ALT allele proportion] = 0.45 was used in a left-sided test. For the *TRAF3* nonsense variant, where the number of reads mapping to the alternative allele was greater than that to the reference allele, an additional test was done to check for significant overexpression of the alternative allele. In that test, we assumed no reference mapping bias and used a null hypothesis of [ALT allele proportion] = 0.5 in a right-sided test.

### 2.5. Exon Usage Analysis

To aid in identifying potential aberrant splice events, exon usage was evaluated in two ways. First, percent spliced in (PSI) was calculated following the method laid out in Schafer et al. [60]. PSI was used to confirm exon skipping events identified in other forms of data. Second, read counts for collapsed exons were obtained following the DEXSeq protocol [61]. To visualize changes in exon usage while controlling for differences in overall gene expression, exons were normalized on a per-gene basis using transcripts per million in order to factor in exon length. These normalized counts were plotted for each gene and used to visually identify genes and specific exons to prioritize for splicing analysis.

### 2.6. Splicing

Splice counts were obtained for annotated and unannotated junctions following a method adapted from Mendelian RNAseq [5]. To ensure that all identified splices were supported by sufficient non-duplicate, uniquely mapping reads, aligned BAM files had duplicate reads removed with Picard Tools and were remapped using STAR after converting back to fastq files. The splice junction output files from STAR were combined and junctions were annotated with the gene name and a list of transcripts that use the junction (for known junctions). To align with established clinical standards for SNP calling, a minimum of 10 non-duplicate uniquely mapped reads were required for an unannotated junction to be further investigated. In addition, these events needed to meet a minimum read support threshold of 10% of the overlapping canonical junction. Events were manually analyzed in IGV viewer to confirm that they were not a result of mis-mapping or a sequencing artefact.

### 2.7. Complementary Analysis

Resulting data files from the above methods were used in a complementary manner to analyze the targeted RNAseq from each patient. After identification of exonic variants, the annotated VCF was used to prioritize genes for manual analysis of the splice counts. Exon usage was used primarily in tandem with spliced read counts to look for aberrant splicing events, but also to identify possible differential isoform usage. Allele-specific expression was calculated with read counts of high-quality exonic variants. Gene expression was evaluated with DESeq2 only in conjunction with truncating variants to support conclusions about nonsense-mediated decay. Identification and interpretation of variants were informed and influenced by the American College of Medical Genetics (ACMG) guidelines for variant classification [62].

## 3. Results

### 3.1. Development of a Targeted RNAseq Panel for Immunodeficiency Analysis

We performed targeted RNA sequencing of 260 genes to an average depth of 11.5 M read pairs for 13 patients suspected of, or known to have, an immune-related disorder. A comprehensive analysis of the RNA was completed for each sample, which included variant calling, identification of aberrant splicing, and outlier gene expression. A summary of findings is given in Table 1 (variants classified as pathogenic/likely pathogenic by ACMG guidelines in bold), and the workflow is outlined in Figure 1.

Our targeted sequencing panel consisted of 260 genes, 104 of which have been implicated in primary immune deficiencies (PIDs) and 194 in very early-onset inflammatory bowel disease (VEOIBD), with 38 overlapping both disease classes. GC content and gene-body coverage were consistent across all samples. An average of 91% of reads in each sample mapped to the targeted genes, with 196 genes (75%) receiving at least 20 mapped reads in every sample. Gene coverage is summarized in Appendix A. In order to determine if the coverage extended across the entire gene, however, it is important to look at splice junction coverage. A recent paper developed a calculation for the minimum read sequencing depth (MRSD) needed to express a given gene or genes at a level sufficient for splicing [63]. Because we did not have a PBMC whole mRNA control dataset for direct comparison of MRSD with our targeted panel, we used the MRSD web tool and combined the results for whole blood and lymphoblastoid cell lines (LCLs), acknowledging that a handful of genes were likely to be specific to a given biotype. MRSD indicated that 88 genes (out of 258 panel genes with splice junctions, 34%) would have greater than or equal to 20 reads mapped to at least 75% of splice junctions in 99% of samples at a sequencing depth <= 50 M reads per sample. With targeted RNAseq, we found that 140 genes (54%) met the same criteria at an average sequencing depth of ~22 M reads (not to mention the fact that we used only non-duplicate reads, which were ~6 M per sample). An additional 19 genes have 50% of exons covered, and if we reduce the confidence interval to 75% of samples (since our cohort is disease samples only and we would expect more variation), we end up with a total of 171 genes (66%) with good coverage for a robust splicing analysis.

### 3.2. Application to Seven Cases of Primary Immunodeficiency

We used our targeted RNAseq approach to evaluate likely causal mechanisms for seven cases of primary immunodeficiency being treated at Emory University clinics. Three of the cases we consider to be resolved on the basis of exonic mutations (sample IDs P25, P49, P69), all of which were previously noted from WES. In each of these cases, RNA provided additional evidence in support of pathogenicity (sample IDs P49, P69) or of potential contributing factors (sample ID P25). Altered therapeutic intervention is suggested for two cases (sample IDs P49, P69). In two other cases (sample IDs P55, P89), RNAseq identified candidate genes and/or variants for further investigation. Digenic inheritance may be implicated in two cases (sample IDs P25, P89). This left just two cases (sample IDs P31, P35) completely unresolved and with no new information.

Patient P69 presented with refractory IBD and a history of recurrent fevers. Exome sequencing revealed a hemizygous deletion encompassing exons 4 and 5 of the *XIAP* (HGNC:592) gene, on the basis of which the patient was started on Anakinra [64], an inhibitor of the IL-1 receptor, with the goal of proceeding to a curative bone marrow transplantation (BMT). Analysis of the RNA provided functional evidence that the deletion results in a shift in the reading frame predicted to lead to a premature stop and a protein assay confirmed XIAP deficiency (Appendix A). Loss of function (LOF) variants in *XIAP* are known to be causative for X-Linked lymphoproliferative syndrome 2 (XLP2, OMIM 300635) [65]. To our knowledge, this is the first time this variant has been reported in a patient with a confirmed XIAP deficiency.

Symptoms in patient P49 included immune cytopenia, IBD, and eczema, and exome sequencing identified a single heterozygous truncating SNV in *CTLA4* (c.442C>T, p.Gln148*; HGNC:2505). This variant has not previously been reported, but other truncating variants in *CTLA4* are known to be causative for autoimmune lymphoproliferative syndrome type 5 (ALPS5, OMIM 616100) by way of haploinsufficiency. Analysis of RNA revealed allele-specific expression occurring in *CTLA4* (Figure 2A) that is suggestive of nonsense-mediated decay (nmd) likely to lead to reduced protein expression. The patient was started on Abatacept [66], a CTLA-4 fusion protein that binds to CD80/CD86 and inhibits T-cell activation, as well as the immunosuppressant sirolimus (rapamycin) [67,68], and subsequently underwent bone marrow transplantation. Two other family members were found to harbor the same variant with only mild symptoms, a common finding among families with *CTLA4* LOF variants that suggests other factors may be involved in disease severity [23]. One explanation is compensation by the wild-type allele, which could potentially be observed by targeted RNAseq of relatives in addition to the proband.

Patient P25 was previously found to have a dominant negative mutation in *CARD11* (HGNC:16393) causing severe atopic disease (IMD11B, OMIM 617638) [69]. Noting the possibility of other variants contributing to disease heterogeneity and severity, analysis of RNA continued and identified a pathogenic missense SNV in *MEFV* (HGNC:6998) along with an intron retention event. Although a second variant was not identified in *MEFV*, the highly penetrant M680I mutation has been previously observed in symptomatic carriers of familial Mediterranean fever (FMF, OMIM 249100) [70,71,72]. Whether or not a carrier with this variant will be symptomatic does not appear consistent within families, suggesting low penetrance and variable expressivity, possibly due to the presence of modifier mutations in other genes. An intron retention event was observed in this patient at the end of *TCF25* (HGNC:29181) intron 9. No causative variant was identified, but due to allele-specific expression and increased usage of the remaining *TCF25* exons (Figure 3), one possible explanation is a larger duplication. Whole genome sequencing or CNV analysis may shed additional light on this event. *TCF25* is important in transcriptional activity involved in heart development and disease [73]. Disruption of *TCF25* could potentially be an additional susceptibility factor in FMF carriers with a highly penetrant variant.

Patient P55 was found to have rare exonic variants in both *NCF1* (c.269G>A; p.R90H; HGNC:7660) and *NCF2* (c.812A>G; p.Lys271Arg; HGNC:7661), which are primarily linked to chronic granulomatous disease (CGD, OMIM 233700). In the homozygous state, *NCF1* R90H has been associated with a case of pediatric interferonopathy [74]. Splicing and exon usage analysis in this patient suggest the presence of an additional pathogenic event in this gene involving exons 2–3. Unfortunately, the high similarity of pseudogene *NCF1B* (HGNC:32522) makes identifying the specifics of this event difficult in RNA alone [75]. Further studies would be needed to confirm whether the mutations in *NCF1* are in trans and hence whether compound heterozygosity explains causation. In any case, the rare events seen in *NCF1* and *NCF2* strongly point towards one or both of these genes being involved in the pathogenesis of this case. In addition, a non-frameshift deletion of a single amino acid was identified in *WAS* (HGNC: 12731), which is associated with Wiskott–Aldrich syndrome (WAS, OMIM 301000). While pathogenicity of this variant has not been determined, we note that carriers of *WAS* pathogenic variants escape disease through non-random X inactivation and that random X inactivation has been found in symptomatic carriers [76,77], presenting the potential for this to be a second or interacting cause.

Patient P89 also harbors a variant in *NCF1* (p.G83R) which has been previously found to reduce reactive oxygen species and is associated with more severe disease course in pediatric IBD [78]. In addition, though no aberrant splicing was detected, abnormal exon usage was observed in the ubiquitin-modifying enzyme (UME) *USP4* (HGNC:12627). UMEs are involved in the regulation of the disease course of IBD [79,80]. Four other rare exonic or UTR variants were identified in other panel genes for this patient. Exceedingly rare variants have been shown in previous research to be overrepresented in early-onset IBD and primary immune patients [81,82], suggesting a complex multigenic disease origin for this case.

### 3.3. Application to Six Cases of Very Early-Onset Inflammatory Bowel Disease

We also used our targeted RNAseq approach to evaluate likely causal mechanisms for six cases of very early-onset IBD being treated at Boston Children’s Hospital. Suggestive genetic abnormalities were detected in all cases, though follow-up assays would be needed to confirm several of these. One case we consider to be resolved on the basis of exonic mutations (sample ID CHB974). Two cases (sample IDs CHB749, CHB535) harbor single variants highly likely to explain disease, though are not yet definitively resolved, while another two cases suggest digenic inheritance (sample ID CHB786) or provide evidence for oligogenic inheritance (sample IDCHB953). Two cases (sample IDs CHB974, CHB786) have NOD2 involvement and three (sample IDs CHB535, CHB1025, CHB953) have variants of interest that influence splicing or transcript abundance. Altered therapeutic intervention is suggested for one case (sample IDCHB749).

*NOD2* (HGNC:5331) has been repeatedly associated with IBD [83,84,85]. A recent study suggested that compound heterozygosity of known *NOD2* risk alleles explains up to 10% of pediatric IBD in European-ancestry cases [86]. In our cohort, patients CHB974 and CHB786 were found to harbor the p.G908R variant. A second *NOD2* risk allele, p.L1007fs, was identified in CHB974, confirming that *NOD2* loss-of-function is likely the causative mechanism in this child. While there were no additional *NOD2* variants found in CHB786, ultra-rare variants were observed in three other genes, including the *NOD2* inhibitor *ERBIN* (HGNC:15842), indicating this case could be digenic.

X-linked agammaglobulinemia (XLA, OMIM 300755), which is characterized by low B-cell counts and is associated with early-onset colitis, is caused by defects in the *BTK* (HGNC:1133) gene. CHB749 was found to be hemizygous for a missense variant (c.1955T>C; p.L652P) in the tyrosine kinase domain of *BTK*. This variant has been reported in patients with XLA previously [87,88], but researchers studying *BTK* variant effects on protein have drawn attention to dissimilarities between this variant and other pathogenic *BTK* variants—while the majority of disease-causing *BTK* variants are missense changes in structurally important residues of the tyrosine kinase domain, L652P is not a well-conserved location and the residue is exposed in the assembled protein structure [89,90]. However, the change to proline in this portion of the kinase domain C-lobe breaks the α-helix, making this change more likely to be disruptive to the protein. In the previous studies that reported the L652P variant in a patient with XLA, protein expression was not determined. Assays to assess BTK protein expression and B-cell levels should be performed in this patient to confirm a diagnosis of XLA. Since intravenous immunoglobulin to treat infections may not improve inflammation from colitis in patients with BTK defects [91], a more creative and personalized treatment plan may be required for this case.

Patient CHB535 was found to contain a nonsense variant in *TRAF3* (c.1275C>G; p.Y425X, HGNC:12033). The extensive functions and interactions of *TRAF3* are still being elucidated, but it is known to be important to inflammatory pathway signaling and gene abnormalities have been associated with many diseases including herpes simplex encephalitis (IIAE5, OMIM 614849), Waldenstrom macroglobulinemia (WM, OMIM 153600), and IBD. The Y425X variant occurs in the highly conserved TRAF-C subunit of the TRAF domain, which is responsible for receptor binding and participates in stabilization of TRAF3 trimerization [92]. Few truncating germline variants have ever been reported in *TRAF3*, and it is not known whether truncating variants are disease-causing. A missense variant was previously found to have a dominant-negative effect on the protein via destabilization of TRAF3 trimers leading to protein expression of only 17.5% compared to the wild-type [93]. This and other studies have shown that deletion of the TRAF-C domain (the predicted effect of the Y425X truncating variant) does not have the same effect and produces the 30% of protein necessary to maintain normal signaling function (an amount that suggests simple haploinsufficiency is not disease-causing) [94,95,96,97]. However, the specific deletion variant created in these studies removed not just the TRAF-C subunit, but also all or part of the TRAF-N subunit shown to be essential for TRAF3 trimerization. Other studies into TRAF3 protein interactions that deleted specifically the TRAF-C subunit in the course of their research unquestionably prove that removal of this domain disrupts inflammatory pathway signaling [98,99], but did not study the wild-type/deletion variant combination. This leaves open the possibility that the Y425X variant observed in CHB535 could act in a dominant-negative manner to cause inflammatory disease. Our RNA analysis provides an important clue to clarifying the pathogenicity of this variant. In order to act in a dominant-negative fashion, the truncated transcript needs to elude degradation by the nonsense-mediated decay (nmd) machinery. This often happens when the truncation occurs in the last coding exon of the transcript [100]. The RNA analysis showed no reduction in transcripts containing the variant (Figure 2B) as well as normal overall expression of *TRAF3*, confirming escape from nmd. Interestingly, the proportion of reads containing the nonsense variant suggest an overexpression of the alternative allele (REF = 286 reads, ALT = 360 reads; *p* = 0.00202). In order to show that TRAF3 protein function is sufficiently deficient and declare Y425X likely pathogenic, a protein assay should be performed.

*MERTK* (HGNC:7027) signaling is important in the negative regulation of inflammation [101]. Two samples, CHB1025 and P69, harbor a missense variant at the end of *MERTK* exon 5 (c.844G>A; p.A282T). This variant has previously been reported in patients with multiple myeloma [102], but in silico predictors do not agree on whether this change would be damaging to protein function, and gnomAD [103] frequency in African Americans is 14%. It is no wonder, then, that submissions of this variant to ClinVar have interpreted it as benign. However, our analysis of mRNA shows this to be a “leaky” variant, where the reduced affinity for the canonical splice site results in a non-frameshift exon skip (Figure 4). Exon 5 of *MERTK* is part of one of the immunoglobulin-like domains that are important for ligand binding in the inflammatory pathway [104]. Only about 15% of total *MERTK* transcripts are mis-spliced, making it unlikely to be a disease-causative mutation for retinitis pigmentosa, the rare disease typically associated with the gene, especially given its high frequency in the African population. The possibility remains, though, that this is a risk allele for immune disorders, in combination with *XIAP* hemizygosity in P69, and an as yet unidentified cofactor in this case of VEOIBD.

Common variable immune deficiency (CVID, OMIM 607594), a primary immune deficiency that has been associated with VEOIBD [31], has been suggested to be polygenic in origin rather than the traditional monogenic mode of congenital disease inheritance. A total of four potentially disease-causing variants were identified in CHB953, three of which are located in genes linked to CVID [105]. A heterozygous nonsense variant was found in *PIK3CD* (HGNC:8977). While PIK3CD-related disease is primarily caused by missense gain-of-function variants, at least two studies have identified loss-of-function variants to be disease-causing as well [106,107]. The second variant identified was a 15 bp deletion encompassing the exon 7 splice donor site of *TYK2* (HGNC:12440). Analysis of RNA showed the resulting mRNA change to also be a non-frameshift deletion of 15 bp, thanks to an alternate splice donor site conveniently located at the beginning of the genomic deletion. Despite this minimal disruption, the deletion removes a portion of the FERM domain, which has been shown to be important to TYK2 protein function [108]. Thirdly, extended splice site variants were found in *UNC13D* (c.154-8T>A; HGNC:23147) and *CAT* (c.903+5G>T; HGNC:1516), both resulting in intron inclusion (Figure 5). The *UNC13D* variant has not previously been reported in the literature but has been reported to ClinVar and interpreted as benign and likely benign. The catalase variant was reported as causative for acatalasemia/hypocatalasemia (OMIM 614097) in a study that found a reduction in catalase levels in patients carrying the variant [109]. Again, this variant has been reported to ClinVar, with interpretations of benign and uncertain significance. Both variants have likely been disregarded due to their frequency—while generally rare, they occur in just over 1% of South Asians according to gnomAD. Usage of the nearby canonical splice donor site in *CAT* is roughly 60% of expected, while canonical splicing in *UNC13D* is just 40% of expected for the affected location. No alternative splicing was observed, and a high number of reads mapped to both introns. In both genes, the intron inclusion is expected to result in premature truncation. *CAT* mRNA levels were not reduced compared to other samples, so a protein assay would be needed to confirm hypocatalasemia. A reduction of *UNC13D* was similarly not seen in the mRNA. However, exonic SNPs across the entirety of the gene were observed in a roughly 40/60 ratio. Coupled with the ~40% use of the canonical splice site, it appears as though rather than nmd this transcript is exhibiting increased expression. While these variants cannot be classified as pathogenic based on the data in this study, we recommend that they be considered VUSs until further research can be done on their effects. Construction of mouse strains with combinations of mutants might reveal the oligogenic basis of the pathogenesis in this individual [110].

## 4. Discussion

This study introduces three innovations with respect to personalized genomic medicine: (i) use of targeted RNAseq to increase the resolution of splicing dysregulation, (ii) development of a modified bioinformatics pipeline for diagnostic evaluation, and (iii) application to a pilot study of 13 cases with two classes of immunodeficiency. Exome sequencing alone had achieved molecular diagnosis of three cases (sample IDs P25, P49, P69) by ACMG guidelines. Using our targeted RNAseq approach, we confirm the likely mechanism of pathology for these three individuals, provide evidence for previously un-noticed mechanisms of disease for another three individuals, and provide suggestive evidence for di- or trigenic inheritance in two more. Although these five additional cases do not meet strict ACMG guidelines for being resolved, we argue that the additional evidence is helpful in resolving candidate genes that further experiments would confirm to ACMG standards. The two major limitations of the approach are that it may only be applicable to immune diseases where common blood cell types are involved, and the targeted RNAseq panel may not include the causal gene in some cases.

The decision to use a targeted panel rather than sequencing the entire transcriptome is unusual, but is validated by evidence that it increases the proportion of splice sites with sufficient read depth to evaluate dysregulation. One of the largest limitations of RNAseq for rare disease diagnostics is that the ability to capture a variant is dependent on that gene’s expression level in the sequenced tissue type. This generally leads to arguments that the disease-relevant tissue is a necessity for RNAseq and/or that sequencing depth should be at least 50–100 M reads per sample [5,38,39]. The recent minimum read sequencing depth (MRSD) study identified whole blood (over LCL, cultured fibroblasts, and skeletal muscle) as the worst option for most gene panels [63]. However, we show that our targeted gene panel outperformed expectations and allowed us to analyze at least 20% more genes of interest than we would have been able to with whole mRNA.

Three variants reported in ClinVar as benign or likely benign were shown to affect splicing in mRNA, drawing the previous interpretation into question. This exemplifies some of the biggest drawbacks to rare variant interpretation in DNA: the dependence on variant frequency and in silico predictors. Splice prediction tools, while useful for narrowing in on variants, will never be as accurate as directly assessing the effect through RNAseq. In addition, assays showing a reduction in functional protein function are interpreted more readily in the context of RNA evidence of the specific change resulting from a splice variant. Variant frequency, while it is (and should) remain a primary way of prioritizing putatively pathogenic variants, should sometimes also be used with some caution where the variant is not necessarily causal, but likely facilitative of dysregulation. Rare disease prevalence is widely thought to be underestimated and is complicated by heterogeneous phenotypes, digenic and polygenic inheritance, and differences between subpopulations [111,112]. For example, acatalasemia/hypocatalasemia prevalence has been estimated at over 2% in some Asian populations [113,114], making it important to consider individual ancestry when interpreting catalase variants. In a polygenic inheritance model, it is possible that a specific combination of variants that are each individually more common than the disease prevalence together create the disease-causative effect. These nuances are increasingly important as we improve the field of personalized medicine to better understand and treat complex rare disease cases.

The method in Cummings et al. has been criticized for lacking a statistical basis and arbitrarily choosing cutoff thresholds [48,115,116]. Since publication, multiple tools for a robust statistical analysis for the identification of aberrant splicing have been developed, most notably FRASER [115]. However, the FRASER paper acknowledges that sample size affects the ability of the tool to detect all known splice events, which highlights the important point that rare disease RNAseq analysis tends to involve small sample sizes. Since our work most closely resembles the Mendelian RNAseq method [5], after completing our splicing analysis we also ran FRASER for comparison. Out of seven splice events manually identified through the original analysis, FRASER detected just two with FDR < 0.1. The FRASER paper suggests that z-score and delta PSI be prioritized over p-value, especially in small cohorts such as ours. Another three events were successfully identified by FRASER using their suggested delta PSI cutoff of 0.3. When the delta PSI threshold was lowered to 0.1 and the read counts were used to prioritize events (a method quite similar to this study and Mendelian RNAseq [5]), FRASER detected six of seven splice events as well as two additional events not found in the original analysis.

For research labs that lack the funds and resources required for whole mRNA sequencing of 100 rare disease samples at a depth of >50 M reads per sample, RNAseq analysis is not at all out of reach. With a well-curated panel, targeted RNAseq can nearly double the number of genes that can be analyzed, at a fraction of the total sequencing depth. A smaller number of genes to analyze means that each individual patient sample can undergo a more thorough analysis that combines variant calling, exon usage information, and identification of splice events using both FRASER and the manual methods used in this study and Cummings et al. to achieve the highest resolution.

We show the potential of the approach to increase diagnostic yield, but much work needs to be done to incorporate findings of this nature into the ACMG guidelines for clinical diagnosis, and thence to improve patient care. We stress that the RNAseq findings regarding variants in this paper do not meet the threshold for categorization as pathogenic or likely pathogenic. All individuals, even those who are considered healthy, contain many rare variants in disease-related genes. However, in the quest to elucidate the genetic causes of rare disease and increase diagnostic yield, the field must look beyond a simple monogenic mode of inheritance. As we learn more about how variants that are not exceedingly rare (but less common than most polymorphisms) contribute to rare disease severity and heterogeneity, it will likely become necessary for additional or adapted guidelines to be developed in order to standardize how we interpret these variants in the context of a patient’s personal variant profile.

## Figures and Tables

**Figure 1 jpm-12-00919-f001:**
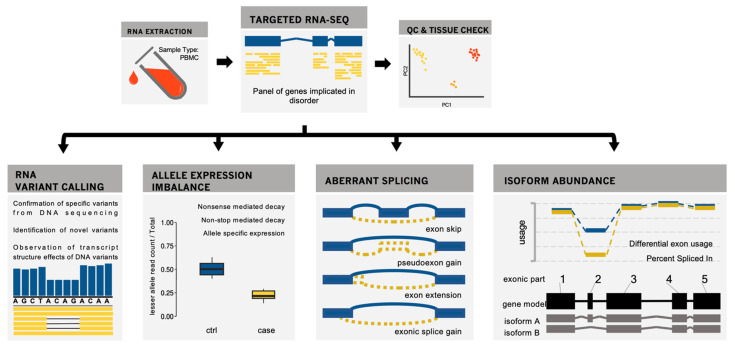
An overview to the RNAseq analysis approach. The visuals in this figure are representative of concepts and do not show data from this study. PBMCs are extracted from whole blood and sequenced using a targeted gene panel. After QC, variants are called in the RNA to confirm variants from exome sequencing and identify variants not captured in exome sequencing. Allele expression imbalance, aberrant splicing, and isoform abundance are analyzed in tandem with called variants to provide a more complete picture of the functional effects of variants on mRNA structure.

**Figure 2 jpm-12-00919-f002:**
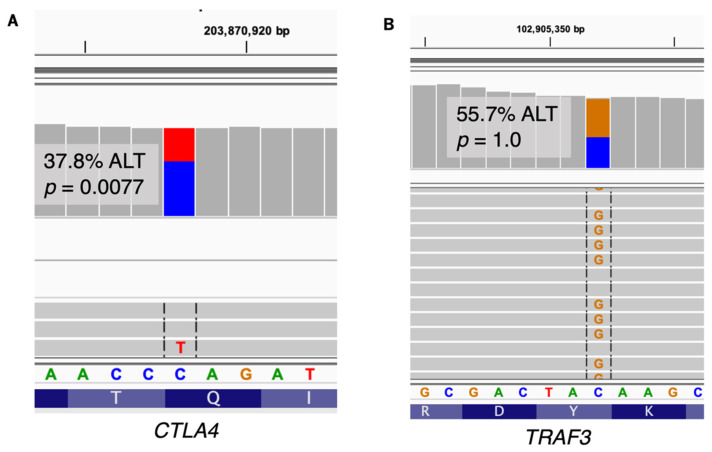
Variant allele-specific expression. (**A**) The nonsense variant in CTLA4 for patient P49 shows unbalanced expression (181 REF, 110 ALT; *p* = 0.0077), suggesting nonsense-mediated decay. (**B**) The nonsense variant in *TRAF3* for patient CHB535 does not exhibit allele-specific expression (286 REF, 360 ALT; *p* = 1.0), suggesting it escapes nonsense-mediated decay. Reference allele (REF) and alternate allele (ALT) read counts refer to allele counts in the final VCF file.

**Figure 3 jpm-12-00919-f003:**
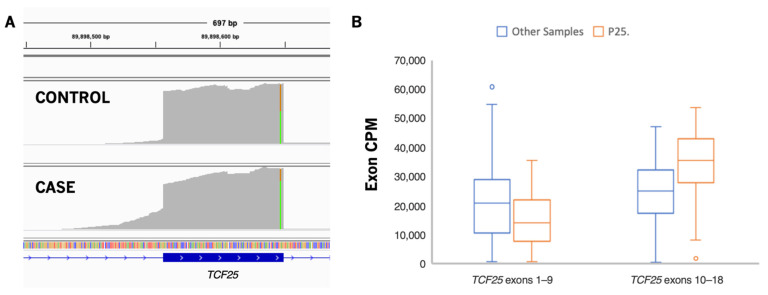
Possible aberrant event in *TCF25*. (**A**) Exon 10 of *TCF25* in an unaffected sample and in P25 showing evidence of intron retention and allele-specific expression. The variant seen in this exon is a common polymorphism, but shows unbalanced allele expression in P25. (**B**) Normalized exon usage in exons prior to the intron retention event compared to the affected exon and following exons.

**Figure 4 jpm-12-00919-f004:**
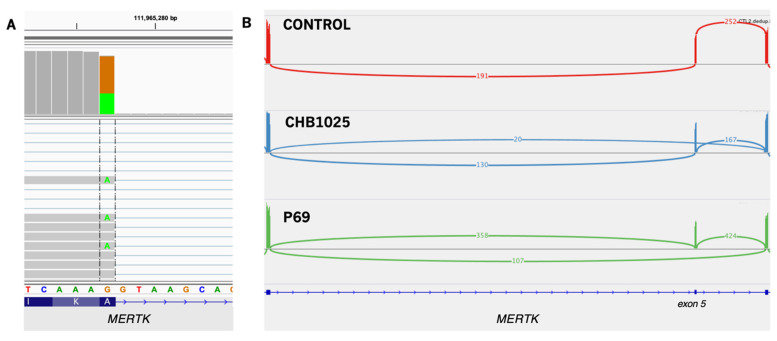
“Leaky” exon skip in *MERTK.* (**A**) The A282T variant is visible at the end of exon 5. About 33% of reads mapping to this location have the G>A change and splice normally. (**B**) Sashimi plot showing the skip of exon 5 in patients CHB1025 and P69. About 15% of total spliced reads skip exon 5. Sashimi plot shows only the affected and immediately adjacent exons.

**Figure 5 jpm-12-00919-f005:**
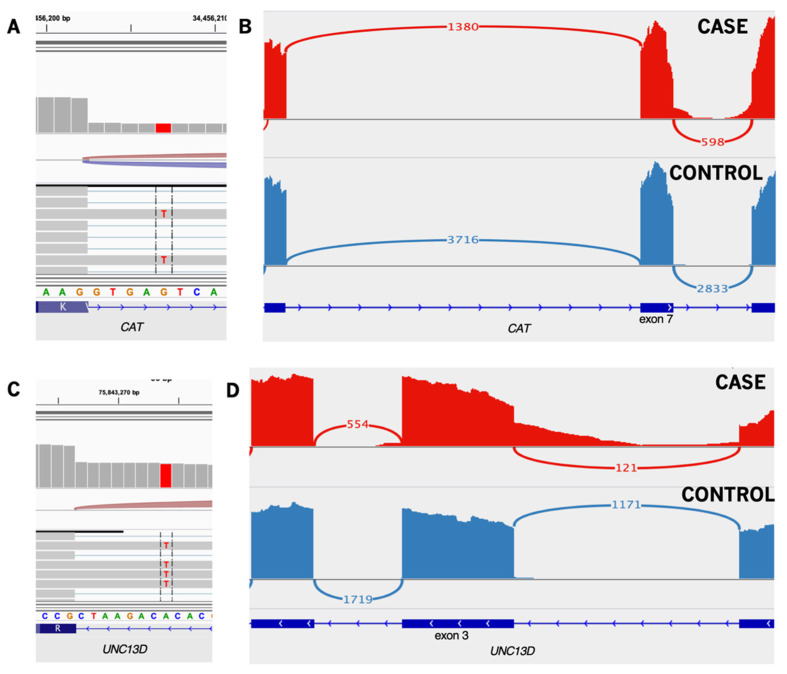
Intron retention events in CHB953. (**A**) The extended splice variant in CAT is present in all unspliced reads. (**B**) Unspliced reads are visible in the CAT intron. Normalized spliced read counts from this exon are around 60% of what is seen in other samples. (**C**) The extended splice variant in UNC13D is present in all unspliced reads. (**D**) Unspliced reads are visible in the UNC13D intron. Normalized splice read counts from this exon are around 40% of what is seen in other samples. Irregular coverage of retained intron reads is due to decay with distance from the pulled-down exon. Sashimi plots show only the affected and immediately adjacent exons.

**Table 1 jpm-12-00919-t001:** Variants and splicing events of interest.

Sample	Gene	HGVS	Predicted AA Change	Variant Type	Zygosity	Splice Effect in RNA	rs no.	SIFT	PolyPhen2	CADD Phred	Interpro Domain	GnomAD MAF
P25	*MEFV*	c.2040G>C	p.M680I	missense	HET	-	rs28940580	Tolerated	Benign	0.002	SPRY	0.0001
	** *CARD11* **	**c.224G>A**	**p.R75Q**	**missense**	**HET**	**-**	**rs1064795280**	**Damaging**	**Prob.Dam.**	**33**	**CARD**	**-**
	*TCF25*			unknown	HET	intron retention	-	-	-	-	-	-
P49	** *CTLA4* **	**c.442C>T**	**p.Q148X**	**nonsense**	**HET**	**-**	**-**	**-**	**-**	**38**	**Ig-like**	**-**
P55	*NCF2*	c.812A>G	p.K271R	missense	HET	-	-	Tolerated	Prob.Dam.	24.4	SH3	-
	*NCF1*	c.269G>A	p.R90H	missense	HET	-	rs201802880	Damaging	Benign	25	Phox homolog	0.001
	*WAS*	c.689AGA{2}	p.K232del	nonframeshift del	HET	-	rs782409127	-	-	-	-	9.67 × 10^−5^
P69	** *XIAP* **			**gross del**	**HEMI**	**x4-5 skip**	**-**	**-**	**-**	**-**	**-**	
	*MERTK*	c.844G>A	p.A282T	missense/ splice	HET	leaky exon skip	rs7588635	Damaging	Benign	24.2	Ig-like	0.0108 (AFR: 0.14)
P89	*NCF1*	c.247G>A	p.G83R	missense	HET	-	rs139225348	Tolerated	Poss.Dam.	20.2	Phox homolog	0.0089
CHB535	*TRAF3*	c.1275C>G	p.Y425X	nonsense	HET	-	-	-	-	35	MATH/TRAF	-
CHB749	*BTK*	c.1955T>C	p.L652P	missense	HEMI	-	rs128622212	Tolerated	Poss.Dam.	20.6	Protein kinase	-
CHB786	*NOD2*	c.2722G>C	p.G908R	missense	HET	-	rs2066845	Damaging	Prob.Dam.	31	Leu-rich repeat	0.0113
	*ERBIN*	c.3704A>C	p.Y1235S	missense	HET	-	rs201900105	Tolerated	Benign	19.43	-	1.39 × 10^−5^
	*RBCK1*	c.992C>T	p.S331L	missense	HET	-	rs781592121	Damaging	Benign	26.3	Zinc finger	9.37 × 10^−5^
	*TYK2*	c.2456G>A	p.S819N	missense	HET	-	rs763006605	Tolerated	Benign	7.356	Protein kinase	5.03 × 10^−5^
CHB953	*PIK3CD*	c.1595delG	p.W532X	nonsense	HET	-	-	-	-	-	-	-
	*CAT*	c.903+5G>T	-	splice	HET	intron retention	rs147912187	-	-	-	-	0.0029 (SAS: 0.01)
	*UNC13D*	c.154-8T>A	-	splice	HET	intron retention	rs369433391	-	-	-	-	0.0027 (SAS: 0.01)
	*TYK2*	c.1001_1011+4del	p.V333_E337del	nonframeshift del	HET	-	-	-	-	-	FERM	2.63 × 10^−5^
CHB974	*NOD2*	c.3017dupC	p.A1006fs	frameshift ins	HET	-	rs2066847	-	-	-	-	0.0151
	*NOD2*	c.2722G>C	p.G908R	missense	HET	-	rs2066845	Damaging	Prob.Dam.	31	Leu-rich repeat	0.0113
CHB1025	*MERTK*	c.844G>A	p.A282T	missense/splice	HET	leaky exon skip	rs7588635	Damaging	Benign	24.2	Ig-like	0.0108 (AFR: 0.14)

## Data Availability

The data presented in this study are available on request from the corresponding author. The data are not publicly available due to protected privacy concerns for participants. Code for generating RNAseq analysis files is available at https://github.com/kiera-gt/rnaseq-pid-veoibd.

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
