# Peer review of "Targeted RNAseq Improves Clinical Diagnosis of Very Early-Onset Pediatric Immune Dysregulation"

_jpm, 2022, doi:10.3390/jpm12060919_

Round 1

Reviewer 1 Report

This study applied targeted RNA-sequencing for the evaluation of the slicing dysregulation in primary immunodeficiencies (PID) and very early-onset inflammatory bowel disease (VEOIBD), which provides a diagnostic approach for these diseases.

Minor:

  1. Page 6-7, Line 222-228. Please add the sample IDs in the descriptions. “Four of the cases (PXX) we consider to be resolved on the basis of exonic mutations, two (PXX) of which were previously noted from WES, whereas the RNA evidence provides additional evidence in support of pathogenicity in two cases. In one (PXX) of these latter cases, digenic inheritance may be implicated on the basis of a retained intron at a second gene, a phenomenon also seen in a fifth case (PXX). This left just two cases (PXX) completely unresolved. Altered therapeutic intervention is suggested for two cases(PXX).”
  2. Page 7, Line 235-236.Please add the reference. Loss of function (LOF) variants in XIAP are known to be causative for X-Linked Lymphoproliferative Syndrome 2 (XLP2, OMIM 300635).
  3. Page 7, Figure 2, figure legend. Please add REF, reference allele. ALT, alternate allele.
  4. Page 9, Line302-315. Please add the sample IDs in the descriptions. “We also used our targeted RNAseq approach to evaluate likely causal mechanisms for 6 cases of very early onset IBD being treated at Boston Children’s Hospital. Three of the cases we consider to be resolved on the basis of exonic mutations, two (PXX) with NOD2 involvement and one hemizygous (PXX) for a missense variant that the RNAseq supports elevating to likely pathogenic. Two other cases (PXX) have variants of interest that influence splicing or transcript abundance, and the final case (PXX) provides evidence for oligogenic inheritance. Suggestive genetic abnormalities were thus detected in all cases, though follow up assays would be needed to confirm several of these. Altered therapeutic intervention is suggested for two cases.”

Author Response

Reviewer 1
This study applied targeted RNA-sequencing for the evaluation of the slicing dysregulation in primary immunodeficiencies (PID) and very early-onset inflammatory bowel disease (VEOIBD), which provides a diagnostic approach for these diseases.

Reviewer comment:
Page 6-7, Line 222-228. Please add the sample IDs in the descriptions. “Four of the cases (PXX) we consider to be resolved on the basis of exonic mutations, two (PXX) of which were previously noted from WES, whereas the RNA evidence provides additional evidence in support of pathogenicity in two cases. In one (PXX) of these latter cases, digenic inheritance may be implicated on the basis of a retained intron at a second gene, a phenomenon also seen in a fifth case (PXX). This left just two cases (PXX) completely unresolved. Altered therapeutic intervention is suggested for two cases(PXX).”

Response:
We appreciate the reviewer’s desire for clarity on which samples these statements refer to as it led to a revision of this paragraph. In the initial submission, we failed to revise an earlier assessment of which cases were resolved that used more liberal criteria.  The new paragraph better reflects what is presented in the textual presentation of data in our results section.  The updated paragraph is lines 226-234 of the revised manuscript.

“Three of the cases we consider to be resolved on the basis of exonic mutations (P25, P49, P69), all of which were previously noted from WES. In each of these cases, RNA provided additional evidence in support of pathogenicity (P49, P69) or of potential contributing factors (P25). Altered therapeutic intervention is suggested for two cases (P49, P69). In two other cases (P55, P89), RNA-seq identified candidate genes and/or variants for further investigation. Digenic inheritance may be implicated in two cases (P25, P89). This left just two cases (P31, P35) completely unresolved and with no new information.”

Reviewer Comment:
Page 7, Line 235-236.Please add the reference. Loss of function (LOF) variants in XIAP are known to be causative for X-Linked Lymphoproliferative Syndrome 2 (XLP2, OMIM 300635).

Response:

“Loss of function (LOF) variants in XIAP are known to be causative for X-Linked Lymphoproliferative Syndrome 2 (XLP2, OMIM 300635)[115].”

  1. Rigaud S, Fondaneche M-C, Lambert N, Pasquier B, Mateo V, Soulas P, et al. XIAP deficiency in humans causes an X-linked lymphoproliferative syndrome. Nature. 2006;444(7115):110-4.

Reviewer Comment:

Page 7, Figure 2, figure legend. Please add REF, reference allele. ALT, alternate allele.

Response:
The Figure 2 legend has been updated to reflect the meaning of REF and ALT, line 263-264 of the revised manuscript.

“REF (reference allele) and ALT (alternate allele) read counts refer to allele counts in the final VCF file.”

Reviewer Comment:

Page 9, Line302-315. Please add the sample IDs in the descriptions. “We also used our targeted RNAseq approach to evaluate likely causal mechanisms for 6 cases of very early onset IBD being treated at Boston Children’s Hospital. Three of the cases we consider to be resolved on the basis of exonic mutations, two (PXX) with NOD2 involvement and one hemizygous (PXX) for a missense variant that the RNAseq supports elevating to likely pathogenic. Two other cases (PXX) have variants of interest that influence splicing or transcript abundance, and the final case (PXX) provides evidence for oligogenic inheritance. Suggestive genetic abnormalities were thus detected in all cases, though follow up assays would be needed to confirm several of these. Altered therapeutic intervention is suggested for two cases.”

Response:

We again thank the reviewer for this comment as we have updated the paragraph for better clarity regarding a summary of our findings. As above, the newly revised summary paragraph is in better concordance with the more conservative assessment criteria.  Please see lines 314-322 of the revised manuscript.

“Suggestive genetic abnormalities were detected in all cases, though follow-up assays would be needed to confirm several of these. One case we consider to be resolved on the basis of exonic mutations (CHB974). Two cases (CHB749, CHB535) harbor single variants highly likely to explain disease, though are not yet definitively resolved, while another two cases suggest digenic inheritance (CHB786) or provide evidence for oligogenic inheritance (CHB953). Two cases (CHB974, CHB786) have NOD2 involvement and three (CHB535, CHB1025, CHB953) have variants of interest that influence splicing or transcript abundance. Altered therapeutic intervention is suggested for one case (CHB749).”

Reviewer 2 Report

Berger et al.'s demonstration on the added diagnostic power from RNA-seq in addition to WES is well written and interesting to the field. I particularly commend the authors on a thorough introduction and article which flows very well. While their data demonstrates the ability to utilize targetted RNA-seq to investigate variants, I find that proper controls are lacking to build confidence on the impact of these variants. Particularly, the authors analyze 13 samples and appear to find RNA variants in each case, but without unaffected family members or similar controls they are unable to confirm that these variants contribute to disease phenotype. For example, in lines 248-249, the authors describe a shared CTLA4 variant between the affected and their family members who demonstrate mild symptons. They then suggest that this variant may not be the sole contributor of the phenotype observed. This may be a similar case for variants observed in other patients. If possible, the authors should include an unaffected  (preferably related) control group to discriminate between inheritted non-pathogenic variants and de novo or inherited pathogenic variants.. Other suggested improvements are below:

  • In lines 235 and 242 the authors measure protein expression as validation of observed premature stop codons. However, they do not show this data in any of their main figures or supplmentary. This should be included.
  • In figure 1, "Allele Expression Imbalance" pane, control data is included in the method overview but this control is not listed in the method section. The authors should explain what data is used as a control.
  • Details of the RNA extraction method should be included in the method section (line 115)
  • cDNA synthesis method (if applicable) should be included, e.g. PolyA vs random hexamer synthesis and more information in the library prep parameters, e.g. RNA input, hybridization time and temperature should also be listed (line 116)

Author Response

Reviewer 2

Berger et al.'s demonstration on the added diagnostic power from RNA-seq in addition to WES is well written and interesting to the field. I particularly commend the authors on a thorough introduction and article which flows very well.

Response:

Many thanks for this positive appraisal of our work, we appreciate it.

Reviewer Comment:
While their data demonstrates the ability to utilize targetted RNA-seq to investigate variants, I find that proper controls are lacking to build confidence on the impact of these variants. Particularly, the authors analyze 13 samples and appear to find RNA variants in each case, but without unaffected family members or similar controls they are unable to confirm that these variants contribute to disease phenotype. For example, in lines 248-249, the authors describe a shared CTLA4 variant between the affected and their family members who demonstrate mild symptons. They then suggest that this variant may not be the sole contributor of the phenotype observed. This may be a similar case for variants observed in other patients. If possible, the authors should include an unaffected  (preferably related) control group to discriminate between inheritted non-pathogenic variants and de novo or inherited pathogenic variants..

Response:
We agree that using a control of unaffected family members would be an excellent and indeed necessary way to identify de novo variants or inherited variant combinations that contribute to differing phenotypes. Unfortunately, we did not have access to family exomes in this study, nor did we have samples from them for targeted RNA-seq. Since this is not a clinical study, and no results were returned to participants, we don’t feel the additional data is needed since it does not affect the proof-of-principle nature of the work (note that the decision to administer Abatacept was made without the RNA data).  However, we do feel that we have been cautious throughout in repeatedly stating where additional tests or research would be necessary to conclusively show contribution to disease or even reclassify variants as pathogenic, and appreciate the reviewer’s suggestions in this regard.  Please note that lines 258-9 did state that targeted RNAseq of relatives would provide additional evidence. As the reviewer is, we are sure, aware, the complexity of inheritance patterns and variable penetrance of rare genetic disorders means that for rare variants, the presence or absence in unaffected family members or other groups is just one piece of evidence amongst many. We expect that this may be even more complicated with regards to digenic and oligogenic inheritance, which we noted both in the results section when suggesting multi-genic contributions as well as in the discussion section where it is one of our main points.

Reviewer Comment:
In lines 235 and 242 the authors measure protein expression as validation of observed premature stop codons. However, they do not show this data in any of their main figures or supplmentary. This should be included.

Response:
The protein expression data for XIAP has been added as a supplementary figure and a reference to it has been added to the text (line 241 of the revised manuscript). We are thankful to the reviewer for this comment as we are committed to ensuring the data we present is as complete and accurate as possible. Though we believed CTLA4 protein expression had been measured, this was a miscommunication between the authors. We have removed the mention of protein expression confirmation in the revised manuscript (lines 249-251).

“Analysis of RNA revealed allele specific expression occurring in CTLA4 (Figure 2A) that is suggestive of nonsense mediated decay (nmd) likely to lead to reduced protein expression.”

Reviewer Comment:

In figure 1, "Allele Expression Imbalance" pane, control data is included in the method overview but this control is not listed in the method section. The authors should explain what data is used as a control.

Response:

The images in Figure 1 are intended as an aid for the reader in conceptually visualizing the types of analyses performed and were not created using real data. There is more than one way to analyze and present Allele Expression Imbalance and the image is simply representative of the concept. The specific method of Allele Expression Imbalance analysis, in this case using probability testing on a single variant to infer the presence of nonsense-mediated decay, is explained in the methods and does not involve use of a control sample. To address the source of confusion, we have added the following statement to the Figure 1 legend for clarity (lines 198-199 of revised manuscript):

“The visuals in this figure are representative of concepts and do not show data from this study.”

Reviewer Comment:

Details of the RNA extraction method should be included in the method section (line 115). cDNA synthesis method (if applicable) should be included, e.g. PolyA vs random hexamer synthesis and more information in the library prep parameters, e.g. RNA input, hybridization time and temperature should also be listed (line 116)

Response:

We thank the reviewer for noting the incomplete description, and have added additional details regarding RNA extraction and sequencing (lines 115-127 of revised manuscript). The specific details of hybridization time and temperature are found in the publicly available protocol for library preparation mentioned and linked in the updated paragraph.

“This study used RNA (median RIN=9.4, range=6.9-9.8) from peripheral blood monocytes (PBMC) as the sample type. PBMCs were extracted from whole blood using Stem cell Technologies Lymphoprep Density Gradient Medium. For our PID samples, RNA was extracted from PBMCs and sent to us directly from Emory University. For our VEOIBD samples, RNA was extracted from PBMCs using the NucleoSpin RNA XS kit (Macherey-Nagel, Germany). For Targeted RNA Seq strand specific library preparation, RNA input per sample was 300ng with target enrichment using a custom Agilent SureSelect Targeted Enrichment panel of 260 genes linked to VEOIBD or PID. All samples underwent Poly-A selection, followed by 150bp paired-end sequencing on an Illumina NextSeq platform performed by Georgia Tech's High Throughput DNA Sequencing Core at an average sequencing depth of 11.5M read pairs. Further details of library preparation can be found in the SureSelectXT RNA Target Enrichment System protocol (https://www.agilent.com/cs/library/usermanuals/Public/G9691-90000.pdf).”

Reviewer 3 Report

Berger and colleagues screened a small cohort of patients with immune-related disorders by targeted RNA-sequencing of peripheral blood monocyte RNAs in order to implement the genetic diagnosis of these rare diseases, in addition to WGS or WES approaches. They found both exonic and splice variants. Two points:

- Regarding the pathogenicity of the variants described in this work, could the authors apply the American College of Medical Genetics (ACMG) classification to these variants and distinguish between VUS, pathogenic and benign variants according to this classification in Table1?

- Why does the intron retention shown in Fig.5 for patient CHB953 present an irregular coverage of reads along the intron? Do the authors have an explanation for this?

Author Response

Reviewer 3

Berger and colleagues screened a small cohort of patients with immune-related disorders by targeted RNA-sequencing of peripheral blood monocyte RNAs in order to implement the genetic diagnosis of these rare diseases, in addition to WGS or WES approaches. They found both exonic and splice variants.

Reviewer Comment:
Regarding the pathogenicity of the variants described in this work, could the authors apply the American College of Medical Genetics (ACMG) classification to these variants and distinguish between VUS, pathogenic and benign variants according to this classification in Table1?

Response:
We appreciate the reviewer’s suggestion of applying ACMG variant classification guidelines – though we frequently mention interpretation of variants in the text, we realize we had not clarified that we were following ACMG guidelines. We have added a statement in the methods to reflect this (lines 184-186 of revised manuscript), and modified the first paragraph of the Discussion (lines 445-453).

“Identification and interpretation of variants was informed and influenced by the American College of Medical Genetics (ACMG) guidelines for variant classification[116].” And

“Exome sequencing alone had achieved molecular diagnosis of 3 cases (P25, P49, P69) by ACMG Guidelines.  Using our targeted RNAseq approach, we confirm the likely mechanism of pathology for these 3 individuals, we provide evidence for previously un-noticed mechanisms of disease for another 3 individuals, and provide suggestive evidence for di- or tri-genic inheritance in two more.  Although these 5 additional cases do not meet strict ACMG guidelines for being resolved, we argue that the additional evidence is helpful in resolving candidate genes that further experiments would confirm to ACMG standards.”

We also highlighted the 3 variants in Table 1 classified as pathogenic or likely pathogenic by ACMG guidelines.  For all variants listed in Table 1, we have also given common criteria used in variant classification such as variant type, in-silico prediction of impact, and population frequency. For all variants with an interpretation other than VUS, we have discussed the evidence in the text, providing a more complete picture. Although strict ACMG criteria remain to be met with, for example, protein assays, the RNAseq analysis in 5 additional cases provides compelling support for such efforts.  Please note that the immunodeficiencies affecting patients in this study (PID and VEOIBD) may not follow a traditional model of simple Mendelian disease inheritance. One of the major points in our paper is that personalized medicine is increasingly leading to the determination that multiple genes and variants may be contributing to disease phenotype. For this reason, though we greatly respect the ACMG guidelines and sought to follow the underlying principles, we are not sure they are best optimized for cases such as these (which we noted in the discussion). We feel that we have been very clear regarding our limitations in the text whenever discussing specific variants where our evidence does not meet the threshold for reclassification of a variant.

Reviewer Comment:
Why does the intron retention shown in Fig.5 for patient CHB953 present an irregular coverage of reads along the intron? Do the authors have an explanation for this?

Response:
Targeted RNA-seq probes were designed based on known exon coordinates. In most instances of a read mapping to an intronic sequence, it was able to capture this only because a portion at the beginning and/or end of the read maps to an exonic sequence (which the probe targeted). This means that in the case of intron retention, intronic coverage will be highest immediately adjacent to an exon and will gradually drop off along a distance of 150bp into the intron (our read length). Irregular coverage in genomic sequencing is common and expected, but we have added a statement to the Figure 5 legend to clarify this (line 439 of revised manuscript).

 “Irregular coverage of retained intron reads is due to decay with distance from the pulled-down exon.”